# Adjuvant rituximab, a potential treatment for the young patient with Graves' hyperthyroidism (RiGD): study protocol for a single-arm, single-stage, phase II trial

Michael Cole,[1] Ann Marie Hynes,[2] Denise Howel,[1] Lesley Hall,[2] Mario Abinun,[3] Amit Allahabadia,[4] Timothy Barrett,[5] Kristien Boelaert,[6] Amanda J Drake,[7] Paul Dimitri,[8] Jeremy Kirk,[9] Nicola Zammitt,[10] Simon Pearce,[11] Tim Cheetham[12]

For numbered affiliations see end of article.

**Correspondence to**
Dr Tim Cheetham;
tim.cheetham@nuth.nhs.uk

## ABSTRACT

**Introduction** Graves' disease (Graves' hyperthyroidism) is a challenging condition for the young person and their family. The excess thyroid hormone generated by autoimmune stimulation of the thyroid stimulating hormone receptor on the thyroid gland can have a profound impact on well-being. Managing the young person with Graves' hyperthyroidism is more difficult than in older people because the side effects of conventional treatment are more significant in this age group and because the disease tends not to resolve spontaneously in the short to medium term. New immunomodulatory agents are available and the anti-B cell monoclonal antibody rituximab is of particular interest because it targets cells that manufacture the antibodies that stimulate the thyroid gland in Graves'.

**Methods and analysis** The trial aims to establish whether the combination of a single dose of rituximab (500 mg) and a 12-month course of antithyroid drug (usually carbimazole) can result in a meaningful increase in the proportion of patients in remission at 2 years, the primary endpoint. A single-stage, phase II A'Hern design is used. 27 patients aged 12–20 years with newly presenting Graves' hyperthyroidism will be recruited. Markers of immune function, including lymphocyte numbers and antibody levels (total and specific), will be collected regularly throughout the trial.

**Discussion** The trial will determine whether the immunomodulatory medication, rituximab, will facilitate remission above and beyond that observed with antithyroid drug alone. A meaningful increase in the expected proportion of young patients entering remission when managed according to the trial protocol will justify consideration of a phase III trial. Ethics and dissemination The trial has received a favourable ethical opinion (North East - Tyne and Wear South Research Ethics Committee, reference 16/NE/0253, EudraCT number 2016-000209-35). The results of this trial will be distributed at international endocrine meetings, in the peer-reviewed literature and via patient support groups.

**Trial registration number** ISRCTN20381716.

### Strengths and limitations of this study

► This is a group of patients in whom current therapy does not usually result in disease resolution, with only 20%–30% remitting after a 2-year course of antithyroid drug treatment with carbimazole; hence there is a significant unmet need.
► The behaviour of the disease in the young patient with Graves' hyperthyroidism in terms of likelihood of remission following antithyroid (thionamide) drug is consistent between reported studies This will help us to comment on the potential impact of the trial intervention in this exploratory trial without studying a large number of patients.
► We will be looking at a range of markers of immune function which may help to establish some of the factors that predict response to intervention in this group of patients.
► It is possible that there is an immunomodulatory effect of rituximab that will not be detected because the trial duration of 2 years is too short.
► The likelihood of remission may be different in a 12-year-old patient with Graves' hyperthyroidism versus a 20-year-old patient, and this consideration has not been factored into the trial design.

## INTRODUCTION

Graves' hyperthyroidism, an autoimmune disorder, has an annual incidence of 1 in 10 000 adolescents (~700 per year) in the UK.[1] The standard first-line treatment is the antithyroid drug (ATD) carbimazole (CBZ), which prevents the thyroid gland from manufacturing thyroid hormone and has an immunomodulatory effect.[2] While CBZ will render most patients biochemically euthyroid in appropriate doses, only 50% of adults will remit following a standard 2-year course of ATD. The proportion of children and adolescents entering remission is considerably

smaller at around 25%, and yet the side effects of CBZ are more prevalent in the young, with 20% experiencing adverse events that range from relatively minor problems such as rashes through to potentially life-threatening agranulocytosis.[3 4] Establishing a euthyroid state can be difficult in the growing person, made more difficult by poor medication concordance in some young people.[5 6] Avoiding relapse close to key life events such as examinations can result in prolonged courses of ATD therapy. Most young people will ultimately require thyroid gland excision (total thyroidectomy) or thyroid gland ablation with radioiodine (RI), but these interventions may be associated with additional risks in the young person and do not represent a cure because the patient is then dependent on lifelong levothyroxine replacement.[7 8] Hence, there is a pressing need to develop interventions that can cure a disease that can have major lifelong implications.[8]

Modern immunomodulatory agents have the potential to ameliorate or 'switch off' the immune response to produce durable remission in patients with Graves' hyperthyroidism. Rituximab (RTX), a chimeric anti-B cell monoclonal antibody (MAb) targeting the surface molecule CD20, leads to reductions in B lymphocyte populations lasting for around 6 months in more than 95% of people following one or two doses.[9] CD20 is only expressed on pre-B lymphocytes and mature B cells; it is not expressed on B lymphocyte stem cells or the committed plasma cell. Circulating B lymphocytes are bound by the antibody and removed via antibody-dependent cellular cytotoxicity, complement-mediated lysis and apoptosis. RTX has been successfully used to treat B cell malignancies, with more than one million patients treated worldwide in recent years.[10] It has also been used in a range of autoimmune disorders because of its favourable safety profile in adults and children.[11 12]

RTX has recognised disease-modifying activity both in people with organ-specific and non-organ specific immune-mediated disorders.[11–17] B cell depletion with RTX has been shown to have disease-modifying effects in rheumatoid arthritis, systemic lupus erythematosus (SLE), autoimmune thrombocytopaenia, myasthenia gravis, and polyangiitis with granulomatosis including classical T-cell-mediated autoimmune diseases such as multiple sclerosis and type 1 diabetes. B cell depletion at sites of active inflammation reduces antigen presentation to autoreactive T lymphocytes, leading to amelioration of the cytotoxic autoimmune attack. While autoreactive B cells may return in some patients following RTX,[18] the impact on disease outcome is not closely linked to circulating antibody levels, and the diverse function of B cells including their impact on T cell activity may explain why there may be an effect of RTX many years postadministration.[19–22]

B lymphocyte depleting immunotherapy appears to have disease-modifying activity in adults with Graves' hyperthyroidism; RTX administration has been associated with encouraging remission rates, with approximately 50% of thyrotoxic adults in case reports and case series becoming euthyroid following treatment with RTX.[23–28] Many of these patients were selected on the basis of more severe disease with relapsed Graves' hyperthyroidism or aggressive orbitopathy. The management of cases where there was no clear response to RTX was characterised by early intervention with ATD, surgery or RI within the first year after RTX therapy, which may have obscured a beneficial effect of immunotherapy on thyroid status. Graves' hyperthyroidism and Graves' orbitopathy (GO) have several immunopathogenic features in common, including shared autoantigens, and several studies indicate that RTX has the potential to modify the natural history of thyroid eye disease.[29–31] However, this has not been a consistent observation, with one randomised trial showing no impact of RTX on GO.[32] The difference in trial outcome has been reviewed,[33] and the differences could reflect factors such as the younger age of those recruited to the trial in Italy.[30]

RTX has an excellent safety record in the young and has been used extensively in paediatric practice to treat autoimmune cytopaenias, juvenile dermatomyositis, juvenile idiopathic arthritis, SLE and renal disorders, with significant side effects uncommon.[16 34] Rates of infection appear to be low.[14] Hypogammaglobulinaemia is a possible but very rare side effect in young people and may reflect an underlying but previously unrecognised immunodeficiency.[35] The rare complication, progressive multifocal leucoencephalopathy, has been seen almost exclusively in adult patients with complex disorders who have received multiple immune suppressants (eg, those with B cell malignancies, multiorgan refractory SLE).

The aim of this clinical trial is to explore whether RTX administration will have a beneficial impact on the disease course in young people presenting with Graves' hyperthyroidism. Specifically, we have set out to establish whether a single 500 mg dose of RTX, when administered together with a 12-month course of ATD, will increase remission rates in young people with Graves' hyperthyroidism.

Previous treatment regimens in adults with Graves' disease or GO have typically used between 1000 and 2000 mg RTX administered in divided doses. In contrast, a single dose of 500 mg RTX was used in a recent trial of patients with GO.[30] The investigators in this trial subsequently reduced the dose of RTX to 100 mg in some of their trial patients because complete peripheral B cell depletion was still seen with this smaller dose. Our selection of 500 mg of RTX was a pragmatic decision bearing in mind the established efficacy in many disorders of 2×1000 mg doses versus the observation that a profound effect on B cell numbers was still observed with a much smaller dose. We were also keen to reduce the complexity associated with repeated visits to hospital for an intravenous infusion. By using a modest dose of 500 mg RTX on one occasion, we anticipated a meaningful effect on immune system function but a reduction in potential side effects.[27]

A course of ATD will be also prescribed in addition to a single dose of RTX so that patients are rendered

clinically and biochemically euthyroid while the immune modulating effects of RTX alter the underlying immune disease process. A dose-titration ATD regimen will be used to reduce the likelihood of a subject having to stop ATD medication while still hyperthyroid.[36] ATD will be stopped before 12 months has elapsed if patients become hypothyroid while on the smallest dose of ATD (5 mg CBZ alternate days). As well as preventing thyroid hormone production, ATD has a specific immunomodulatory effect, including reducing thyroid autoantibody concentrations during treatment.[37 38] These effects may involve alteration of thyroid antigen structure,[2] inhibition of proinflammatory cytokines or inhibition of T lymphocytes by other potential mechanisms.[39] An additive or indeed synergistic effect of RTX and ATD is theoretically possible via combined effects on autoantibody generation, and the combination of RTX and ATD may increase remission rates above those observed in studies to date. Another advantage of the immunomodulatory approach to autoimmune hyperthyroidism is the potential for intervention to reduce the likelihood of longer term hypothyroidism.[40 41]

## METHODS AND ANALYSIS
### Trial objectives
The primary objective is to establish whether a single 500 mg dose of RTX, when administered together with a 12-month course of ATD, is likely to result in a meaningful improvement in the proportion of young people with Graves' hyperthyroidism entering disease remission.

The secondary objectives are to examine (1) the relationship between thyrotropin receptor antibody (TRAb) titre and thyroid hormone status (thyroid stimulating hormone [TSH], free thyroxine [FT4] and free tri-iodothyronine [FT3]) in recruited subjects at the beginning and end of the trial; (2) whether immune cellular response is related to disease outcome by examining the relationship between time to recovery of B cell numbers measured in peripheral blood to within the local normal reference range and thyroid hormone status at the end of the trial; (3) whether total dose of ATD is related to disease outcome; (4) the time taken for TSH concentrations to normalise post-RTX and to describe thyroid status, as assessed by TSH, FT4 and FT3 concentrations (normal or abnormal), in the period between cessation of ATD therapy and the end of the trial in each patient; and (5) the safety of the trial treatment regimen by determining the nature and frequency of adverse events.

### Trial design
This is an investigator-initiated, open-label, single-arm, single-stage, phase II trial using an A'Hern design[42] to determine whether the proposed new treatment is likely to meet a minimum level of efficacy before comparing it with standard treatment in a randomised trial. A single-arm design has been chosen because use of a control arm would require a much larger sample size resulting in a longer, more expensive trial when the potential benefit of RTX is, as yet, unknown. The design requires specification of the smallest remission rate, which if true would clearly imply that the treatment warrants further investigation, and the largest remission rate below which we would not wish to proceed to a larger definitive trial.

### Trial setting
This is a multicentre trial based in seven paediatric and seven adult tertiary endocrine units in the cities of Birmingham, Edinburgh, Newcastle, Leeds, Sheffield, Cardiff and Southampton UK and one secondary paediatric unit in Doncaster, UK.

### Inclusion criteria
► Excess thyroid hormone concentrations at diagnosis: elevated free FT3 and/or FT4 (based on local assay).
► Suppressed (unrecordable) TSH (based on local assay).
► Patients between the ages of 12 and 20 years inclusive who are less than 6 weeks from the initiation of ATD treatment (CBZ or propylthiouracil [PTU]) for the first time.
► Elevated thyroid binding inhibitory immunoglobulin or thyroid receptor antibodies (TRAb including thyroid binding inhibiting immunoglobulin [TBII]) based on local assay. Patients may or may not have a raised thyroid peroxidase antibody titre.
► All patients must be willing to use effective forms of contraception for 12 months post-treatment with RTX.
► If women are of childbearing potential, they must have a negative pregnancy test at screening. This will need to be repeated on the day of RTX administration if more than 7 days has elapsed since the screening visit or a negative pregnancy test.
► Able and willing to adhere to a 2-year trial period.

### Exclusion criteria
► Previous episodes of autoimmune thyroid disease.
► Patients with an active, severe infection (eg, tuberculosis, sepsis and opportunistic infections) or severely immunocompromised patients.
► Patients with known allergy or contraindication to CBZ and PTU.
► Participants with previous use of immunosuppressive or cytotoxic drugs (including RTX and methylprednisolone, but excluding inhaled glucocorticoid and oral glucocorticoid for asthma or topical glucocorticoid for eczema).
► Chromosomal disorders known to be associated with an increased risk of autoimmune thyroid disease, including Down's syndrome and Turner syndrome.
► Pregnancy, planned pregnancy during the trial period or current breast feeding.
► Absence of informed consent from parent/legal guardian for participants age <16 years.

- ► Participants with significant chronic cardiac, respiratory or renal disorder or non-autoimmune liver disease.
- ► Participants with known allergy or contraindication to RTX or methylprednisolone.
- ► Participants with evidence of hepatitis B/C infection, assessed by determining hepatitis 'B' surface antigen (HBsAg) status, hepatitis 'B' Core antibody (HB Core antibody) status and hepatitis 'C' virus antibody (HCV antibody) status.
- ► Participants in families who know they will be moving out of the catchment areas during the 2 years following RTX treatment.
- ► Participants currently involved in any other clinical trial of an investigational medicinal product (IMP) or who have taken an IMP within 30 days prior to trial entry.

## Patient identification

Potentially eligible patients will be referred by their local clinician (general practitioner, paediatrician, physician or endocrinologist) or through self-referral via national organisations such as the British Thyroid Foundation (BTF) to a clinician with delegated responsibility to discuss the trial.

## Screening, recruitment and consent

Informed consent discussions will be undertaken by appropriate site staff, with an opportunity for the patient and the parent/guardian to ask questions. Following receipt of the patient information sheet (PIS), participants will be given a minimum of 24 hours to decide whether or not they would like to participate. As part of the consent discussions, the requirement to use effective contraception must be discussed. The PIS states that after the patient has given consent, they will be required to provide a blood sample and undergo a pregnancy test (if female) to confirm that they are not pregnant and that there is no evidence of hepatitis B/C infection. Some sites will also need to carry out HIV screening and QuantiFERON tests to confirm that the patients do not have tuberculosis and HIV (as per their local clinical policies) before a patient can be confirmed as eligible. Those wishing to take part will provide written informed consent/assent prior to trial-specific procedures/investigations. The information sheets and consent/assent forms are shown in online supplementary appendix 1 a, b, c and d.

If thyroid antibody status (TRAb or TBII) was not determined at diagnosis, then this will be assessed at this consent and screening visit. The patient will be classified as a screen fail if they are found not to be eligible at this point. Ineligible patients will be notified via telephone and not asked to come in for a further trial visit. A screening log will be kept securely at each of the recruiting units (paediatric and adult units) to document details of eligible patients who have screen failed or declined to participate in the trial, including any reasons available.

The right to refuse to participate without giving reasons will be respected.

## Intervention

RTX will be administered as a 500 mg dose by slow intravenous infusion under cover with methylprednisolone 125 mg intravenously, chlorpheniramine 10 mg intravenously, and 500 mg (12–16 years of age) or 1 g (>16 years) of paracetamol (orally). The RTX infusion will be started at a rate of 50 mg/hour, increasing by 50 mg/hour every half hour if tolerated; otherwise the infusion will either be maintained at the current rate or stopped and then restarted after 30 min. If the RTX infusion is tolerated and the rate increased as above, then the infusion will take 3.25 hours. Symptoms which are considered intolerable are angioedema, low blood pressure and difficulty breathing; if any are observed, the treatment will be stopped immediately, only to be restarted once symptoms have resolved. Vital signs including blood pressure will be measured every 30 min during the RTX infusion and for 2 hours afterwards.

RTX administered may only take place following confirmation that the patient is hepatitis B/C antibody-negative and within 7 days of a negative test for pregnancy (women). Administration of live vaccines is contraindicated or not recommended for at least 6 months following RTX administration.

## ATD therapy (CBZ and PTU)

A 12-month course of ATD will be administered to induce a euthyroid state in the months following diagnosis. Therapy will normally be with CBZ, which can be administered once daily. PTU is a second-line therapy because of the increased likelihood of hepatic failure with this drug. It is proposed that CBZ dose be titrated against prevailing thyroid function tests in order to maintain a euthyroid state. Clinicians will be encouraged to follow the schedule outlined in table 1 when administering ATD to patients enrolled into this clinical trial, although the managing clinician can elect not to follow the above framework if their assessment of the overall clinical picture suggests that this is in the patients' best interests. PTU can be administered on the basis that 5 mg of CBZ is equivalent to 50 mg PTU. If patients become hypothyroid (with an elevated TSH and low FT4/FT3 level) on the smallest daily dose of CBZ (5 mg) or PTU (50 mg), then the ATD can be administered on alternate days. If patients are still hypothyroid, then the ATD will be stopped before 12 months has elapsed following RTX therapy.

Any participants who develop unacceptable toxicity to CBZ (notably neutropaenia with a neutrophil count less than $0.5 \times 10^9$/L) will stop ATD immediately. Participants with a neutrophil count between 0.5 and 1.0 can remain on ATD if well. The neutrophil count should be repeated after approximately 1–2 weeks or as clinically indicated to confirm that it has not fallen to less than $0.5 \times 10^9$/L. PTU can be commenced as a replacement ATD in the case of adverse events that do not involve hepatic dysfunction or

**Table 1** CBZ dose adjustment according to changing thyroid hormone status

| Week/stage of trial | Thyroid hormone level: FT4 and FT3 (pmol/L) | CBZ dose (mg) |
| --- | --- | --- |
| Presentation | | 20 mg CBZ once daily (consider 10 mg in the case of mild hyperthyroidism, FT3<10). |
| Week 4 | Biochemically hypothyroid (low FT4, FT3 not raised). | Reduce to 5 mg. |
| | FT3 normal and FT4 falling. | Consider reducing from 20 mg to 10 mg, or from 10 to 5 mg. |
| | FT3 elevated. | Continue prior dose. |
| Week 8 | Biochemically hypothyroid (low FT4, FT3 not raised). | Reduce to 20 mg, 10 mg, 5 mg, or reduce CBZ to 5 mg alternate days. |
| | FT3 normal and FT4 falling. | Reduce from 20 mg to 10 mg, from 10 mg to 5 mg, or remain on 5 mg. |
| | FT3 elevated but improving. | Continue prior dose. |
| | FT3 raised or increasing further. | Consider increasing CBZ by 10 mg daily. |
| Week 12 and beyond | Biochemically hypothyroid (low FT4, FT3 not raised). | Reduce to 10 mg, 5 mg or reduce CBZ to 5 mg alternate days. |
| | FT3 normal (or elevated with low FT4). | Continue current dose. |
| | FT3 elevated. | Consider increasing CBZ by 10 mg daily or 5 mg daily if previously euthyroid. |
| Week 16 and beyond | TSH elevation (>4 mU/L) (TSH takes precedence over thyroid hormone levels). | Reduce CBZ dose by 5–10 mg daily. |
| | TSH still suppressed. | Follow guide above for week 12. |

CBZ, carbimazole; FT3, free tri-iodothyronine; FT4, free thyroxine; TSH, thyroid stimulating hormone.

neutropaenia. The increased risk of liver dysfunction on PTU will be discussed with participants and their families prior to commencing PTU. Patients on PTU will have liver function checked at each visit and PTU stopped if ALT or bilirubin is elevated (2× above the upper limit of normal). If patients are unable to tolerate ATD, then it may be feasible to monitor patients off treatment or manage them symptomatically with beta blockade on the basis that RTX may alter the disease natural history and result in remission without further ATD. This option can be discussed with the participant/the participant and their family. Other potential means of maintaining a euthyroid state can be discussed with the trial management team.

Patients who relapse in the second year of the trial will be encouraged to return to ATD in the first instance.

### Outcome measures
#### Primary
The number of subjects in remission at 2 years following a single dose of RTX and a 12-month course of ATD is the primary trial endpoint: this is equivalent to the number who have not relapsed. Subjects will be deemed to have relapsed if they are receiving any concomitant ATD medication in the second year post-RTX administration, or have undergone surgical (thyroidectomy) or RI treatment because of hyperthyroidism at any time following RTX administration. They will also be deemed to have relapsed if serum TSH is less than the lower limit of the normal laboratory reference range and serum FT3 is above the upper limit of the normal laboratory reference range at 2 years.

#### Secondary
1. TRAb titre and related thyroid hormone status at the time of RTX administration and 2 years post-RTX.
2. Time to recovery of B cell lymphocyte numbers in peripheral blood to the normal local lab reference range in relation to thyroid hormone status.
3. Cumulative dose of ATD (mg/kg) in relation to thyroid hormone status 2 years post-RTX.
4. Time taken for TSH and thyroid hormone concentrations to normalise to within the local laboratory reference range post-RTX, and describe biochemical thyroid status in the period between cessation of ATD and 2 years post-RTX in each patient.
5. The frequency and nature of adverse events.

### Visit details and assessments
Following screening and consent, RTX is administered at the baseline visit (week 0). Subsequent visits are scheduled until 24 months; the details are provided in table 2. In addition, after each visit in the first 12 months, participants are contacted by telephone to document changes in ATD dosages. Calls should be completed within 10 days of the visit.

HBsAg, HB Core antibody and HCV antibody status is checked at screening/consent to establish hepatitis status to determine if the patient can be entered into the trial. A full blood count is conducted to look for evidence of

**Table 2** Schedule of events

| Assessments/Intervention | Trial period | | | | | | | | | | | | | | |
| --- | --- | --- | --- | --- | --- | --- | --- | --- | --- | --- | --- | --- | --- | --- | --- |
| | Screening/Consent | Trial intervention | Follow-up | | | | | | | | | | | | |
| | Week –6 to 0 | 0 | 4 | 8 | 12 | 16 | 20 | 28 | 36 | 44 | 52 | 65 | 78 | 91 | 104 |
| Confirmation of eligibility | X | X | | | | | | | | | | | | | |
| Informed consent and assent | X | | | | | | | | | | | | | | |
| HBsAg, HB Core and HCV antibodies | X | | | | | | | | | | | | | | |
| Pregnancy test (women) | X | X | | | | | | | | | X | | | | |
| Medical history | | X | | | | | | | | | | | | | |
| Clinical examination | | X | X | X | X | X | X | X | X | X | X | X | X | X | X |
| Height | | X | | | | | | X | | | X | | X | | X |
| Weight | | X | X | X | X | X | X | X | X | X | X | X | X | X | X |
| RTX infusion | | X | | | | | | | | | | | | | |
| ATD regimen | | X | X | X | X | X | X | X | X | X | X | | | | |
| ATD compliance | | | X | X | X | X | X | X | X | X | X | | | | |
| Concomitant medication | | X | X | X | X | X | X | X | X | X | X | X | X | X | X |
| Adverse and serious adverse event | | X | X | X | X | X | X | X | X | X | X | X | X | X | X |
| Thyroid function (TSH, FT4 and FT3) | X | X | X | X | X | X | X | X | X | X | X | X | X | X | X |
| Liver function (ALT and bilirubin) if on PTU | X | X | X | X | X | X | X | X | X | X | X | X | X | X | X |
| Thyroid antibodies (TRAB and TPO) | X | | | | X | | | X | X | | X | | | | X |
| Lymphocyte subsets* | X | X | | | X | | | X | X | | X | | | | X |
| Full blood count | X | X | X | | X | | | X | X | | X | | | | X |
| Immunoglobulin levels (IgG, IgM, IgA) | X | | | | X | | | X | X | | X | | | | X |
| Specific antibody levels (tetanus, Hib, pneumococcus) | | X | | | | | | | | | X | | | | X |
| Thyroid function (TSH, FT4 and FT3)—central analysis | | X | | | | | | | | | X | | | | X |
| Serum for exploratory analyses | | X | | | | | | | | | X | | | | X |

ATD dose is adjusted at each visit if required.

*Lymphocyte subsets: T cells (CD3), helper T cells (CD4), cytotoxic T cells (CD8), B cells (CD19) and class switch B cells (CD27+ IgD–).

ALT, alanine transaminase; ATD, antithyroid drug; FT3, free tri-iodothyronine; FT4, free thyroxine; HB Core, hepatitis 'B' Core antibody; HBsAg, hepatitis 'B' surface antigen; HCV, hepatitis 'C' virus antibody; Hib, haemophilus influenzae type B; PTU, propylthiouracil; RTX, rituximab; TPO, thyroid peroxidase; TRAb, thyrotropin receptor antibody; TSH, thyroid stimulating hormone.

change in white cell number (a recognised side effect of ATD). Lymphocyte subsets including T cells (CD3+), helper T cells (CD4+), cytotoxic T cells (CD8+), B cells (CD19+) and class switch B cells (CD27+ IgD−) and serum immunoglobulin (Ig) levels (IgG, IgM, IgA) are measured to look at the impact of RTX (anti-B cell MAb) on B and T cell populations and immunoglobulin levels (produced by plasma cells that differentiate from B cells).

Specific vaccine antibodies (against tetanus, haemophilus influenzae type B and pneumococcus) are assessed at baseline and at 12 and 24 months to look for evidence of falling levels as a consequence of the intervention.

At each visit, concomitant medications are recorded, an adverse event check is performed and thyroid function (TSH, FT4 and FT3) is assessed.

We plan to undertake exploratory analyses of immune system function on blood samples obtained prior to the RTX infusion at baseline and then after 12 months and 24 months. This will include measurement of a range of novel factors involved in immune system activity, including B lymphocyte activating factor, a proliferating-inducing ligand and B lymphocyte chemoattractant (CXCL13). These data will be analysed separately from the main trial.

### Withdrawal

Participants have the right to withdraw from the trial at any time, including during the single infusion of RTX, without having to give a reason. The patient will still be asked to complete follow-up. Sites will need to record the amount of RTX that was administered prior to the patients' withdrawal. Investigator sites should try to ascertain the reason for withdrawal and document this reason within the case report form and the participant's medical notes.

As this is a single-dose administration, it is not anticipated that the investigator would withdraw a patient from

treatment (the infusion would be slowed down accordingly—see the Intervention section). If the RTX infusion has to be stopped completely and cannot be recommenced, then those subjects receiving less than 100 mg RTX will be reviewed according to the protocol for safety reasons only. Should a participant withdraw from the trial, then every effort will be made to obtain follow-up data, with the permission of the patient.

Participants who withdraw from the trial will not be replaced.

### Sample size
Formal justification to proceed to a larger randomised trial is based on observing a minimum number achieving remission 2 years post-RTX administration; this number is referred to as the critical number.[42] The critical number depends on the desired and unacceptable remission rates, set at 40% and 20%, respectively, and the error levels set at 10% alpha (type I) and 20% beta (type II). The remission rate corresponding to the critical number will lie between 40% and 20% and will be specified and stored separately in the Statistics trial master file (TMF). If the true remission rate is 40%, there is an 80% chance (power) of proceeding to a further trial; if the true remission rate is 20%, there is a small 10% (alpha) chance of proceeding to a further trial. With these parameters the target recruitment is 27 patients who will receive RTX. This is the smallest number of patients which satisfies the design criteria, assuming a 10% loss to follow-up.

### Statistical analysis
Patients will receive a single dose of RTX once they have fulfilled the inclusion and exclusion criteria. In the event that RTX is not well tolerated, only patients who were able to receive more than 100 mg of RTX will be included in the statistical analysis.

#### Analysis of the primary outcome measure
The number of patients in remission 2 years after RTX administration will be compared with the critical number. This is equivalent to a formal comparison of the hypotheses that the remission rate is greater than or equal to 40% as opposed to less than or equal to 20%. The primary outcome measure, remission rate, will be presented with a one-sided 90% lower bound. If this phase II trial provides evidence that the true remission rate is plausibly 40% or more, 2 years after a single dose of RTX and a 12-month course of ATD, then this will indicate a likely effect of RTX on disease outcome and justify a randomised efficacy evaluation of this adjuvant RTX regimen.

#### Analysis of secondary outcome measures
1. The distribution of TRAb titres will be compared between patients in and out of remission by graphical summary, numerical summary statistics (median and IQR) and a Mann-Whitney test.
2. Time to recovery of peripheral blood B cell lymphocyte numbers (CD19+ cells) to above 70% of the patient's pre-RTX level in relation to thyroid hormone status.

This will be compared between patients in and out of remission by graphical summary, numerical summary statistics (median and IQR) and a Mann-Whitney test.
3. Cumulative dose of ATD (CBZ) in mg/kg in relation to thyroid hormone status. This will be compared between patients in and out of remission by graphical summary, numerical summary statistics (median and IQR) and a Mann-Whitney test.
4. We will assess the time taken for TSH concentrations to normalise (increase above 0.1 mU/L) post-RTX and describe biochemical thyroid status, as assessed by TSH, FT4 and FT3 concentration, in the period between cessation of ATD therapy at the end of year 1 and the end of the trial. Specifically we will plot the time taken for TSH to normalise in individual patients and determine the proportion of time that TSH, FT4 and FT3 concentrations are within the normal reference range in the period between ATD stopping (end of year 1) and trial end (end of year 2).
5. The frequency and nature of adverse events.

#### Criteria for the premature termination of the trial
The criteria for stopping the trial will be if a patient dies from an RTX treatment-related infection, or if two patients experience the same serious unsuspected serious adverse reaction (SUSAR) which has severe or life-threatening consequences.

### Safety reporting
We do not aim to collect adverse events such as minor trauma (scratches, cuts) in patients who are otherwise well. Adverse events will otherwise be classified according to whether they are:
1. Infections (minor: self-limiting infection such as the common cold; modest: non-self-limiting where an antibiotic is prescribed but the clinical course and recovery is not related to the underlying condition and its treatment; or severe: where antibiotic therapy ± other therapy is required with the infection related to drug effects on immune function).
2. Immune function-related such as neutropaenia.
3. Related to the skin, for example, rashes.
4. Related to the musculoskeletal system, such as joint pain.
5. Related to gastrointestinal upset.
6. Related to hepatic function.
7. Others.

The trial will be stopped if a patient dies from an RTX treatment-related infection or if two patients experience the same SUSAR which has severe or life-threatening consequences.

According to section 4.6 of the summary of product characteristics (SmPC) (Mabthera [the product name for RTX] 100 mg and 500 mg concentrate for solution for infusion), RTX should not be administered to pregnant women. The SmPC states that 'Rituximab should not be administered to pregnant women unless the possible benefit outweighs the potential risk'. Due to the long

retention time of RTX in B cell-depleted patients, women of childbearing potential need to use effective contraceptive methods during treatment and for 12 months following RTX therapy (in line with the SmPC for RTX). If a female participant becomes pregnant within 12 months of taking the RTX drug, the details of the pregnancy need to be reported to the chief investigator, trial manager and sponsor within 24 hours of the site learning of its occurrence on the pregnancy reporting form.

If a patient or their partner (where the participant is male) becomes pregnant, within 12 months of taking RTX, the pregnancy must be reported as per the trial-specific guidance document for pregnancy reporting and followed up to completion of pregnancy.

### Patient and public involvement

We were keen to involve patients including those with Graves' hyperthyroidism and their families in the design of this clinical trial. We therefore discussed the proposed trial with patients who had Graves' hyperthyroidism within our clinical service and liaised with the Young Person's Advisory Group (YPAG) in the North of England. We also discussed the trial with the BTF, and it was clear from these discussions that improving the treatment options for young people with this condition was an important area. YPAG and a representative from the BTF helped to refine the trial protocol and the patient information sheets. A member of the BTF was part of the trial steering committee, and the BTF published information regarding the trial on their website and facilitated recruitment by providing information to potential participants.

We intend to notify participants in writing regarding the trial outcome.

### ETHICS AND DISSEMINATION

Trial oversight consists of Trial Steering Committee (TSC), Data Monitoring Committee (DMC) and Trial Management Group (TMG). The TSC comprises an independent chair and three other independent members including a statistician. The DMC consists of three independent members, an endocrinologist (chair), immunologist and a statistician. The TMG will be responsible for overseeing the progress of the trial. The day-to-day management of the trial will be coordinated by the Newcastle Clinical Trials Unit, which will also be responsible for informing sites regarding protocol modifications and updates. Data management will be undertaken in accordance with the Newcastle University Clinical Trials Unit Standard Operating Procedures.

The results of this trial will be distributed widely at international endocrine meetings, including the European Society for Paediatric Endocrinology annual meeting. The results will be published in the peer-reviewed literature and distributed via patient support groups including the BTF.

### DISCUSSION

Graves' disease remains a challenge for young people and their families, with significant disadvantages associated with all standard treatment modalities. Most young people will ultimately be treated with thyroid hormone replacement which is not ideal from a practical, financial and quality of life perspective.[43] The management of Graves' hyperthyroidism is currently under review by National Institute for Health and Care Excellence.

This is a proof-of-concept trial that may be the precursor to a phase III trial. The limitations of the trial design include the need to adopt a largely pragmatic approach to the selection of the RTX dose and the fact that an impact on disease course may not be detected by a 2-year trial. On the other hand, patients who remit by 2 years may still be at risk of relapse at a later stage. There is also a risk that patients will experience side effects from ATD therapy which means this has to be stopped. If so, then definitive treatment may be required before the impact of RTX on the disease course can be elucidated. There is also a need to be cautious about assuming that remission rates following a course of ATD in a 12-year-old patient with Graves' hyperthyroidism are similar to those in someone who is 20 years of age. Remission rate may therefore reflect the age of trial participants as well as the impact of the treatment regimen.

We feel that there are three disadvantages from the patient's perspective when taking part in this trial. First, the infusion involves a day spent in hospital. The second disadvantage is the potential risk of adverse events, notably infections, although the literature does not suggest significant issues in an otherwise healthy and immune competent group of subjects like young patients with Graves' hyperthyroidism who have received one or two doses of RTX. Finally there is the uncertainty about whether the intervention will impact on disease outcome favourably.

The number of clinic visits will be similar to patients with Graves' managed routinely and will help to ensure that patients' biochemistry is monitored carefully.

ATD does not cure young patients with Graves' in the short term and is only associated with resolution of the hyperthyroid state in a minority of patients after several years of treatment. Surgery and RI remove or destroy the gland and hence are not a cure either. RTX provides the prospect of an earlier remission in patients with Graves' hyperthyroidism who are therefore less likely to experience the side effects of ATD. RTX may cure patients who would otherwise not have been cured by ATD, and it may reduce the likelihood of long-term hypothyroidism. Repurposing RTX for this indication would mean an immediately translatable therapy with a well-established and favourable side effect profile, and a likely reduction in costs now that it is off-patent.

## TRIAL STATUS

This manuscript is based on trial protocol V.4.0 dated 27 November 2017. The adjuvant rituximab, a potential treatment for the young patient with graves' hyperthyroidism (RiGD) trial, opened to recruitment in October 2016 and is due to close to recruitment in October 2018.

### Author affiliations

¹Institute of Health and Society, Newcastle University, Newcastle upon Tyne, UK
²Newcastle Clinical Trials Unit, Newcastle University, Newcastle upon Tyne, UK
³Institute of Cellular Medicine, Newcastle University, Great North Children's Hospital, Newcastle upon Tyne, UK
⁴Academic Directorate of Diabetes and Endocrinology, Royal Hallamshire Hospital, Sheffield, UK
⁵C/O Diabetes Unit, Birmingham Children's Hospital, Birmingham, UK
⁶Institute of Metabolism and Systems Research, College of Medical and Dental Sciences, Institute of Biomedical Research, University of Birmingham, Birmingham, UK
⁷Centre for Cardiovascular Science, Queen's Medical Research Institute, Edinburgh, UK
⁸The Academic Unit of Child Health, Sheffield Children's NHS Trust Western Bank, Sheffield, UK
⁹Department of Endocrine, Birmingham Children's Hospital, Birmingham, UK
¹⁰Edinburgh Centre for Endocrinology and Diabetes, Royal Infirmary of Edinburgh, Edinburgh, UK
¹¹Institute of Genetic Medicine, Newcastle University, Newcastle upon Tyne, UK
¹²Department of Paediatric Endocrinology, Institute of Genetic Medicine, Newcastle University, Great North Children's Hospital, Newcastle upon Tyne, UK

**Acknowledgements**  We would like to acknowledge the guidance provided by the local Young Person's Advisory Group and by the British Thyroid Foundation when the trial documentation was being developed. We would also like to acknowledge the ongoing support provided by the clinical and research teams from the endocrine units in Cardiff, Doncaster, Leeds and Southampton which opened as participating sites after the trial commenced.

**Contributors**  SP and TC had the original idea for the trial. The trial protocol was developed by TC, with substantial input and revision by SP, MC, DH, MA and AMH. LH and AJD also made changes to the protocol at various stages of development. Comments from all other coauthors helped to refine the protocol during the final stages leading up to the start of patient recruitment (AA, TB, KB, PD, JK, NZ). The trial protocol manuscript was primarily drafted by MC and TC, with substantial input from DH and MA. All other coauthors approved the final version of the manuscript. All trial coapplicants and investigators who recruit patients to the trial will be eligible for coauthorship.

**Funding**  The trial is funded by a grant from the Medical Research Council Biomedical Catalyst: Developmental Pathway Funding Scheme (DPFS).

**Competing interests**  None declared.

**Patient consent**  Not required.

**Ethics approval**  The trial has received a favourable ethical opinion (North East - Tyne and Wear South Research Ethics Committee, reference 16/NE/0253, EudraCT number 2016-000209-35). REC approval was obtained on 15 September 2016. The first patient was consented on 4 November 2016.

**Provenance and peer review**  Not commissioned; externally peer reviewed.

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
