## [Reviewer comments · BMJ Open]

ARTICLE DETAILS

TITLE (PROVISIONAL)	Adjuvant Rituximab, a potential treatment for the young patient with Graves' hyperthyroidism (RiGD): study protocol for a single arm, single stage, phase II trial
AUTHORS	Cole, Michael; Hynes, Ann Marie; Howel, Denise; Hall, Lesley; Abinun, Mario; Allahabadia, Amit; Barrett, Timothy; Boelaert, K; Drake, Amanda; Dimitri, Paul; Kirk, Jeremy; Zammitt, Nicola; Pearce, Simon; Cheetham, Tim

VERSION 1 – REVIEW

REVIEWER	Hoon Sung Choi Kangwon National University School of Medicine, South Korea
REVIEW RETURNED	03-Aug-2018

GENERAL COMMENTS	There has been a unmet clinical need for refractory Graves' disease, despite of good response to antithyroidal drug. So, I think that immune modulating treatment is promising. However, the subjects of this study is newly diagnosed Graves' disease within 6 weeks. In most cases, Graves' disease shows a good response to antithyroid drugs, so this study setting could be not fit into clinical unmet need. Furthermore, patients will be treated with rituximab and antithyroidal drug. Although this study shows significant beneficial results, this study will not be able to confirm that the cause of the good results is from rituximab, in an open-label, single arm study design.
---

REVIEWER	Brendan C. Stack, Jr., MD, FACS, FACE UAMS, Little Rock, AR, USA
REVIEW RETURNED	03-Aug-2018

GENERAL COMMENTS	A single dose of an immune modulating drug followed by thyroid suppression for a year and withdrawal is intriguing. This could represent a new paradigm for treatment of this and other autoimmune endocrinopathies.
--

REVIEWER	Melvin Khee Shing Leow Tan Tock Seng Hospital, Republic of Singapore
REVIEW RETURNED	21-Aug-2018

GENERAL COMMENTS	The study protocol is quite clear. It is questionable if a slightly increased FT3 above the upper normal limit and a slightly reduced TSH below the lower normal limit at 2 years is an unequivocal relapse of Graves' disease. There are potentially many possible causes of the above, including non-thyroidal illness and medications.
---

VERSION 1 – AUTHOR RESPONSE

Reviewer 1

There has been an unmet clinical need for refractory Graves' disease, despite of good response to antithyroidal drug. So, I think that immune modulating treatment is promising.

However, the subjects of this study is newly diagnosed Graves' disease within 6 weeks. In most cases, Graves' disease shows a good response to antithyroid drugs, so this study setting could be not fit into clinical unmet need. Furthermore, patients will be treated with rituximab and antithyroidal drug. Although this study shows significant beneficial results, this study will not be able to confirm that the cause of the good results is from rituximab, in an open-label, single arm study design.

Thank-you for your useful comments which have provided us with the opportunity to highlight the background to this trial in more detail. The fact that current treatments for Graves' disease are disappointing has been highlighted in the literature in the past and (following this reviewers comments) we have expanded the first point in the 'strength and limitations' section on page 5 to reflect this.

We agree that most young patients will become biochemically euthyroid on antithyroid drug and have included new text on page 6 to reflect this.

However patients are not usually 'cured' because they relapse when the ATD is stopped. Hence young patients are either on antithyroid drug for many years or are ultimately on thyroid hormone replacement following thyroidectomy or radio-iodine therapy. We have highlighted the fact that anti-thyroid drug is more likely to be associated with side-effects in the young, less likely to result in remission and can result in deaths (the introduction on page 6). Hence the limitations of current therapy are particularly pertinent in the young. It is the failure to cure most young patients with Graves' hyperthyroidism that we would regard as an unmet need.

The attraction of the combination of anti-thyroid drug and Rituximab as laid out in this protocol is that it may improve the likelihood of remission rates after only 12 months of therapy. Much longer courses of anti-thyroid drug are typically used.

We would like to highlight reference 8 (an article in the journal, Lancet) where the authors highlight the fact that 'novel approaches to treat the underlying disease process rather than inhibition or destruction of the thyroid' could be a future development. We would also like to highlight reference 3 where the authors state that 'Considering the hepatotoxicity risk associated with PTU, and the other minor and major adverse events associated with both PTU and MMI, strong consideration should be given to the development of less toxic antithyroid medications for use in children and adults'.

We agree with the reviewer that the trial will not be able to confirm that the cause of an apparent beneficial response is definitely due to Rituximab and the statistical design is based on looking for an 'efficacy signal' that will justify proceeding to a randomised control trial. The rationale for the trial design and the possibility of moving onto a larger definitive trial are highlighted in the 'Trial design' section on page 10 and 11 and in the sample size calculation on page 21 and then again in the 'analysis of the primary outcome measure' on page 22. We would also like to highlight the sentence on page 27 in the discussion that states that 'This is a proof of concept trial that may be the precursor to a phase III trial'.

Reviewer: 2

A single dose of an immune modulating drug followed by thyroid suppression for a year and withdrawal is intriguing. This could represent a new paradigm for treatment of this and other autoimmune endocrinopathies.

Thank-you for your encouraging comment.

Reviewer: 3

The study protocol is quite clear.

It is questionable if a slightly increased FT3 above the upper normal limit and a slightly reduced TSH below the lower normal limit at 2 years is an unequivocal relapse of Graves' disease. There are potentially many possible causes of the above, including non-thyroidal illness and medications.

Thank-you for this comment. We did think carefully about the grounds for stating that someone with Graves' hyperthyroidism had relapsed at 2 years. The trial team felt that in someone with established Graves' disease the likelihood of them developing an abnormally low TSH and raised Free T3 at this specific time point (12 months after stopping therapy) by chance or by some other mechanism was remote. We have ensured that measuring TRAB is part of the trial protocol at 2 years to make sure that individuals with a suppressed TSH and raised FT3 also have the pathogenic antibodies associated with relapse.

VERSION 2 – REVIEW

REVIEWER	Hoon Sung Choi Kangwon National University School of Medicine, Republic of Korea
REVIEW RETURNED	05-Oct-2018
GENERAL COMMENTS	The authors answered appropriately the questions.